# Enhancing the Effectiveness of Oligonucleotide Therapeutics Using Cell-Penetrating Peptide Conjugation, Chemical Modification, and Carrier-Based Delivery Strategies

**DOI:** 10.3390/pharmaceutics15041130

**Published:** 2023-04-03

**Authors:** Saeed Anwar, Farin Mir, Toshifumi Yokota

**Affiliations:** Department of Medical Genetics, Faculty of Medicine and Dentistry, University of Alberta, Edmonton, AB T6G 2H7, Canada

**Keywords:** antisense oligonucleotide, viral and non-viral vectors, bioconjugation, cell-targeting moieties, cell-penetrating peptides, nanocarriers, lipid nanoparticles, nucleoside modification

## Abstract

Oligonucleotide-based therapies are a promising approach for treating a wide range of hard-to-treat diseases, particularly genetic and rare diseases. These therapies involve the use of short synthetic sequences of DNA or RNA that can modulate gene expression or inhibit proteins through various mechanisms. Despite the potential of these therapies, a significant barrier to their widespread use is the difficulty in ensuring their uptake by target cells/tissues. Strategies to overcome this challenge include cell-penetrating peptide conjugation, chemical modification, nanoparticle formulation, and the use of endogenous vesicles, spherical nucleic acids, and smart material-based delivery vehicles. This article provides an overview of these strategies and their potential for the efficient delivery of oligonucleotide drugs, as well as the safety and toxicity considerations, regulatory requirements, and challenges in translating these therapies from the laboratory to the clinic.

## 1. Introduction

The rapid advancements in genetics and molecular biology over the past century have opened up new opportunities for treating diseases at the genetic level through genetic therapeutics [1]. Genetic therapeutics involve the use of nucleic acids or their like to modify, control, repair, replace, add, suppress, or delete genetic sequences in the human body for therapeutic, diagnostic, or preventive purposes [2,3]. This is typically achieved by delivering exogenous nucleic acid sequences to the target tissue. Recent developments in biotechnology have enabled powerful technologies for generating and precisely modifying nucleotides and have led to a deeper understanding of biology, which has resulted in nucleic acid-based therapeutics becoming a prominent therapeutic approach among others, e.g., small molecules and protein therapeutics, for treating a wide range of human diseases [3]. These therapeutic nucleic acids can use naturally occurring nucleic acids, e.g., DNA and RNA, their chemical analogs, or a combination of them [4]. Nucleic acid-based therapeutics have the potential to treat various disorders that are currently untreatable through conventional pharmaceutical strategies. The recent development and worldwide use of COVID-19 mRNA vaccines have further demonstrated the capability of nucleic acid-based therapeutics in healthcare [5].

Oligonucleotides (ONs) are synthetic, short strands of nucleic acid analogs that have been designed to target specific RNA sequences [6]. The monomers of an ON are typically chemical analogs of DNA and RNA that have been modified to improve their stability, reduce susceptibility to nuclease degradation, and enhance the specificity and affinity of their binding to the target RNA [7]. ONs can bind to RNA sequences at various stages of their life cycle, including pre-mRNA, mRNA, ribonuclear-protein, or miRNA [6]. The nucleotide sequence of an ON determines the target, while the backbone chemistry determines its pharmacokinetic properties, e.g., stability, solubility, and metabolism. The combinatorial nature of ONs allows for the design and customization of ONs to target any given sequence, providing a significant level of versatility compared to that with conventional pharmaceuticals. Furthermore, the pharmacokinetic characteristics of an ON drug can be adjusted independently, providing a significant level of flexibility in terms of development compared to that with conventional pharmaceuticals [8].

Despite the significant flexibility and applicability of ON therapeutics, several challenges impede their widespread clinical application [9,10,11]. Current research on the design and functional delivery of ONs is focused on developing highly stable ONs with high specificity that can efficiently cross the biological lipid membranes of targeted tissues to achieve the desired modification in a large number of cells. These efforts also aim to ensure optimal immunorecognition outcomes with minimal immune reactions, off-target effects, and adverse effects. Additionally, the scalability of the production of ON drugs at a reasonable and affordable cost is another major challenge that needs to be overcome before ON drugs can be widely adopted in clinical settings.

There has been a continued effort to improve the efficacy of nucleic acid-based therapies through various techniques, e.g., chemical modifications of the ONs, the use of lipid or polymeric nanocarriers, conjugation of ONs to receptor-targeting agents, such as carbohydrates, peptides, or aptamers, and the use of small molecule enhancers. This review provides a comprehensive overview of ON therapeutics, discussing available and emerging strategies to enhance potency, focusing on biological aspects, mechanistic and translational potential, safety and toxicity considerations, and regulatory requirements.

## 2. ON-Based Therapeutic Platforms

Therapeutic ONs are short, synthetic strands of nucleic acid polymers that can be used to modify gene expression through various mechanisms. These include targeting DNA, RNA, proteins, and protein sections, as well as post-translational modifications. Researchers continue to explore, discover, and synthesize new classes of ONs, as well as repurpose and revitalize previously explored classes. However, for the purpose of this write-up, the discussion is limited to the classes depicted in Figure 1, which provides a simplified overview of the mechanisms employed by clinically applicable ON classes.

### 2.1. Gapmers

Gapmers are 16–20-base single-stranded ONs with a central region of 6–10 natural DNA nucleotides and 3–5 sugar-modified ONs on either end [12,13]. The ends are typically composed of locked nucleic acids (LNAs), 2′-O-methyl (2′-OMe), 2′-O-methoxyethyl (2′MOE), or 2′-fluoro (2′-F)-modified bases (Figure 2), enhancing binding affinity, minimizing immune reactions, and increasing nuclease resistance. Gapmers also frequently contain PS bonds throughout, which improve pharmacokinetic characteristics and promote stability and plasma protein binding [12,13,14]. Gapmers, named for the DNA ‘gap’ between the modified ONs, are designed to be complementary to their RNA target, allowing for the formation of a gapmer–RNA duplexes via Watson–Crick base pairing. This gapmer–RNA duplex mimics endogenous RNA–DNA hybrids, which are recognized, cleaved, and eventually degraded via RNase H-mediated cleavage.

Gapmers, historically being the most widely used class of ONs, are naturally taken up by cells in the central nervous system, eye, liver, kidney, adrenal glands, and lungs via endocytosis [12,15]. After being released from endosomes into the cytoplasm, they proceed to the nucleus via an as-yet-unclear mechanism. Gapmers are effective in silencing genes, even in notoriously difficult-to-transfect T-cells, and have shown promise in vivo for gene suppression [16,17]. They target toxic proteins or protein isoforms caused by gain-of-function or dominant-negative mutations [18]. Mipomersen and Inotersen are two FDA-approved gapmer drugs. Mipomersen is indicated for homozygous familial hypercholesterolemia (HoFH) [19], binding to APOB-100 mRNA and degrading it via RNase H1-mediated cleavage, thereby reducing the toxic ApoB-100 level. Inotersen, on the other hand, is indicated for the treatment of hereditary transthyretin amyloidosis (hATTR) and targets the mRNA of transthyretin (TTR) [20], preventing the abnormal extracellular deposition of TTR proteins that give rise to hATTR.

### 2.2. Aptamers

The term “aptamer” is a combination of two Latin root words: aptus, meaning “to fit”, and meros, meaning “part” [21]. Aptamers are single-stranded, chemically modified DNA or RNA molecules that can bind to specific targets, e.g., proteins, peptides, carbohydrates, small molecules, toxins, and even live cells, by fitting to their targets [12]. They range from 20–100 nucleotides in length and can form various shapes due to their ability to form helices and single-stranded loops. They are highly adaptable and can fold into specific 3D architectures to bind their targets, with high selectivity and specificity, and alter the function of the target [12,21,22].

**Figure 2 pharmaceutics-15-01130-f002:**
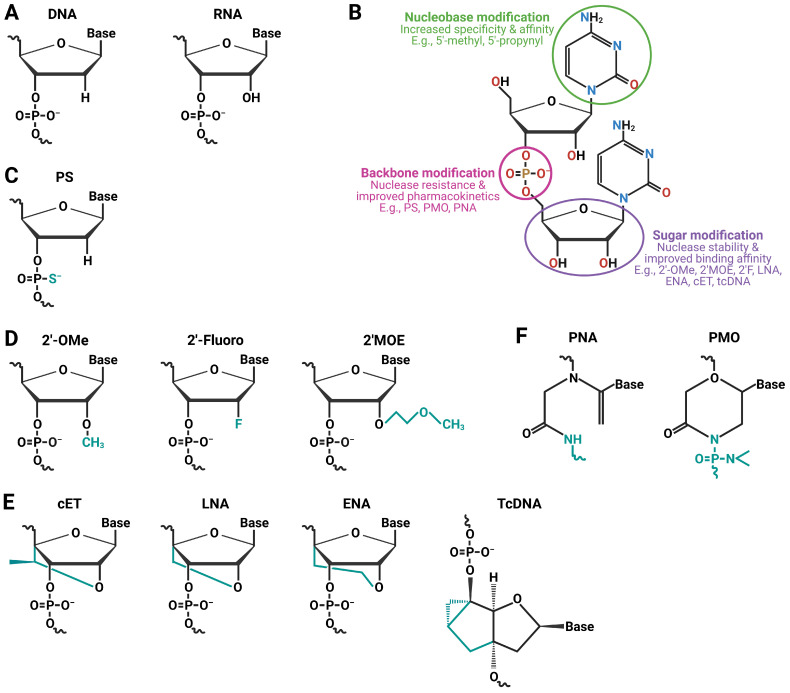
Some commonly used ON chemistries. (**A**) DNA and RNA are two naturally occurring nucleotides. (**B**) Common modification sites on a nucleotide. (**C**) The phosphorothioate (PS) backbone replaces the natural phosphodiester (PO) bond in naturally occurring nucleotides. This enhances the affinity of PS for binding to serum proteins, which helps it evade degradation, causes RNase H to cleave the target RNA, and prevents it from being rapidly excreted by the kidneys [18,23,24]. The structural formula shows a PS-modified DNA monomer. (**D**) The most frequent modifications to ribose at the 2′-O positions of RNA and DNA are 2′-O-methyl (2′-OMe), 2′-O-methoxyethyl (2′MOE), and 2′-fluoro (2′-F). Similar to PS, these changes aid in protecting it from nuclease degradation. These modifications boost the thermal stability of the hybrid that the ON is going to make, thus providing the opportunity to use shorter ONs. However, due to their strong resemblance to RNA rather than DNA, these cannot undergo RNase H cleavage [23]. (**E**) Constrained 2′-O-ethyl (cEt), locked nucleic acid (LNA), 2′-O,4′-C-ethylene-bridged nucleic acids (ENA), and tricyclo DNA (tcDNA) are conformationally constrained analogues to DNA created by putting in a methyl bridge from the 2′-O to 4′ positions of the ribose sugar. These modify the overall charge of the nucleotide and provide flexibility [14,25,26,27]. (**F**) Peptide nucleic acids (PNAs) and phosphorodiamidate morpholino oligomers (PMOs) are charge-neutral nucleotides, the results of nucleobase modifications [28]. PMO has morpholine rings as its backbone joined by phosphorodiamidate linkages; the ribose ring is opened by oxidation then re-closed using ammonia to form a substituted morpholine moiety with phosphorodiamidate linkages substituting phosphodiester bonds. (**C**–**F**) The cyan color shows the modifications made (Created with BioRender: VW2571ZK5I).

Aptamers are often referred to as “chemical antibodies” as they recognize and bind proteins similarly to protein antibodies [29]. In binding targets, they often inhibit protein-protein interactions, eliciting therapeutic effects e.g., antagonism [22]. Unlike other nucleic acid drugs, aptamers are not rationally designed. Rather, they are identified using a technique called systematic evolution of ligands by exponential enrichment (SELEX), a directed in vitro evolution process that partitions large libraries of degenerate ONs for target binding iteratively and alternately [30,31]. They are then enzymatically amplified until the sequencing of cloned individuals reveals functional sequences. Aptamers are truncated, modified at the 2′-O position, capped at their termini to enhance nuclease resistance, and attached to PEG to reduce renal clearance [12,22]. They are chemically synthesized in a scalable process that introduces specific conjugation points with defined stereochemistry.

Although once believed to be immunoinert, aptamers have been found to be capable of inducing immune responses when administered systemically [32]. They have advantages over antibodies, including superior uptake, scalable synthesis, and rapid switching off [12,21,22,33]. Moreover, aptamers can be rapidly “switched off” by introducing a complementary ON strand. Furthermore, because aptamers comprise 2′-O modified ONs, toll-like receptor-mediated innate immune responses are suppressed [22,33].

Pegaptanib sodium, indicated for macular degeneration, was the first aptamer drug to receive approval from the FDA in 2004. However, despite the benefits aptamers show over antibodies, pegaptanib is losing market share to more effective monoclonal antibodies. As a result, aptamers are not currently a competitive therapeutic option for protein antibodies [34]. Nonetheless, new aptamers are being clinically evaluated for various hematology, oncology, ocular, and inflammatory indications [22].

### 2.3. Steric-Blocking ONs

Steric-blocking ONs are synthetic, single-stranded ONs of 15–30 bases length that are chemically modified to enhance nuclease resistance, reduce immune response risks, and increase target RNA binding affinity. Common modifications include PS-modified DNA (PS-DNA), 2′MOE, 2′OMe, LNA, 2′-thio-DNA, PNA, and PMO [27] (Figure 2). Among these, 2′OMe and 2′MOE modifications have been historically the most researched, but newer chemistries are continuously being explored [35,36]. Additionally, peptide and/or polymer-conjugated chemistries are often used [27]. Steric-blocking ONs do not cause RNase H-mediated target RNA degradation; instead, they inhibit transacting factors, e.g., small nuclear RNAs (snRNAs), microRNAs (miRNAs), and long non-coding RNAs (lncRNAs) or prevent RNA secondary structure formation [18,37].

The first application of steric-blocking ONs dates back to the mid-1980s when an ASO targeted a 3′ splice site in herpes simplex virus pre-mRNA, preventing splicing and acting as an antiviral agent [38]. Since then, these ONs have been widely used to correct mutation-driven splicing defects and modulate mRNA stability and protein translation [39]. Steric-blocking ONs are thought to be taken up by cells in the CNS, eye, liver, kidney, adrenal glands, lungs, and, to a lesser extent, skeletal muscle [15]. They are being investigated as therapeutics for genetic diseases and pathogenic microorganisms, e.g., bacteria and viruses [6,27,40,41]. Sarepta Therapeutics’ PMO-based drug eteplirsen received FDA approval to treat Duchenne muscular dystrophy (DMD) amenable to exon 51 skipping, despite the approval process being fraught with controversy [42]. Eteplirsen binds to DMD exon 51 during the pre-mRNA splicing and sterically blocks the inclusion of exon 51 in the final transcript—leading to the production of a truncated but partially functional DMD protein. Between 2019 and 2021, three more PMO-based steric blocking ON drugs, namely golodirsen, viltolarsen, and casimersen, were approved for the treatment of exons 53 and 45 amenable DMD [43,44,45]. Nusinersen, composed of 2′MOE ONs, is another steric blocking ON FDA-approved drug for treating spinal muscular atrophy (SMA) caused by mutations in chromosome 5q that lead to SMN protein deficiency [46]. The first N-of-1 genetic therapeutic, Milasen, was a 2′MOE-based steric blocking ON developed to treat a single patient with neuronal ceroid lipofuscinosis [47]. Our lab has previously reviewed Eteplirsen [48], Golodirsen [49], Viltolarsen [50,51], Casimersen [52], and Nusinersen [53] comprehensively; details about the mechanisms of action, pharmacodynamics, and clinical aspects of these drugs can be read from those reviews.

### 2.4. Immunostimulant ONs (ISOs)

Bacterial DNA can stimulate an immune response due to the presence of unmethylated CpG dinucleotides in specific sequence contexts, known as CpG motifs [54,55,56]. Synthetic single-stranded ONs containing unmethylated CpG motifs, also known as CpG oligodeoxynucleotides (CpG ONs) or immunostimulatory ONs (ISOs), can mimic the immunostimulatory properties of bacterial CpG DNA [57,58]. They are being studied as potential molecular adjuvants for vaccines and as treatments for diseases with an inflammatory component, e.g., cancer, diabetes, amyotrophic lateral sclerosis (ALS), and other conditions, such as asthma and arthritis [58,59,60]. ISOs mimic pathogen-associated molecular patterns (PAMPs), which are molecular markers of pathogens that, when recognized by cells, e.g., antigen-presenting cells, dendritic cells, and B cells, trigger an immune response [59,61]. There are at least three distinct classes of CpG ONs, namely CpG-A, CpG-B, and CpG-C [58,62], which differ in their structural composition. The structural differences and the sequence and chemistry used to synthesize them lead to these classes of CpG ONs triggering different immune responses.

ISOs can augment the induction of antigen-specific adaptive immune responses and promote the development of cytotoxic CD8+ T cells against tumor antigens [57,58,63]. CpG ONs are widely used as therapeutic agents and adjuvants for cancer vaccinations, triggering an immunological response by activating TLR9 [57,58,63]. However, ISOs often face two significant obstacles: their PS-backbone is counterproductive for accumulation in the immune system, and the presence of CpGs makes them inefficient in crossing the cellular membranes [64,65,66]. CpG molecules are unable to be internalized by antigen-presenting cells following parenteral injection due to their small molecular size, highly hydrophilic backbone, and negative charge. To address these issues, CpG ONs are often conjugated with lipids or other molecules to increase their cellular absorption and accumulation in antigen-presenting cells in lymph nodes [59,64].

The first CpG ISO-containing vaccine to receive FDA approval was Heplisav-B, which is indicated for the hepatitis B virus [62,67]. Developed by Dynavax Technologies, Heplisav-B features the hepatitis B surface antigen as the principal component of the vaccine and is adjuvanted with CpG 1018, a 22-mer long PS-modified DNA ON. The vaccine was approved by the FDA in 2017. In comparison to Energix-B (GSK), a three-dose hepatitis B vaccine that uses aluminum hydroxide as an adjuvant, CpG 1018 induced a significantly improved immune response [68]. CpG 1018 has been clinically trialed for COVID-19 vaccines and is being trialed in an HIV vaccine [69,70,71,72].

### 2.5. Antagomirs, RNA Sponges, and Blockmirs

Antagomirs and RNA sponges (ARSs) are both methods used to silence microRNAs (miRNAs) [73,74,75]. Antagomirs, also known as anti-miRs, are synthetic single-stranded nucleic acids (ONs) designed to specifically bind to endogenous miRNAs, whereas RNA sponges bind to multiple miRNAs with the same sequence in their ‘seed region’. As they hybridize with their targets, they sterically block their function. Both antagomirs and RNA sponges are composed of PS-modified 2′OMe and 2′-O-ethyl nucleotides and are often conjugated with lipid or peptide moieties at the 3′ end (Figure 2) [76].

miRNAs bind to their target mRNA molecules and prevent their translation. RNA sponges bind to miRNAs and lead to their degradation through the RNA-induced silencing complex (RISC); as a result, the target mRNA of the miRNA is not prevented from translating its protein product [76]. However, miRNAs are often promiscuous and regulate the expression of several mRNAs, making it challenging to differentiate the specific effects on their intended target versus additional effects on several other mRNAs [77]. Blockmirs, which use the same compositional features as the ARS, bind to miRNA-binding sites on the 3′ untranslated region (UTR) of mRNA, allowing translation to occur [78,79]. As a result, blockmirs are binding-site sequence-specific and not miRNA-specific, making their activity more predictable and less likely to cause off-target effects.

### 2.6. Agomirs

Agomirs are functional miRNA mimics that bind to miRNA-binding regions of target genes, eliciting enhanced repressive effects [80]. Typically, all the nucleotides in an agomir are 2′-OMe or 2′MOE, with the ones at both ends being PS-modified (Figure 2) [80]. Additionally, other modifications in the sugar (e.g., 2′-F, LNA, PNA, and PMO) and backbone (e.g., methylphosphonate and boranophosphate) are being tested [81]. Furthermore, most agomirs are conjugated with peptide, lipid, or biotin moieties at the 3′ end to enhance cellular uptake [80,81]. Endogenous miRNAs are only partially complementary to all their targets and depend on imperfect miRNA–target mRNA binding to elicit their function, making their effect non-specific and promiscuous [77]. However, the sequence of an agomir is strictly complementary to the miRNA-binding site, ensuring target-specific action [80]. Agomirs flood the mRNA inhibition pool and, using the exact mechanism as miRNAs, manifest enhanced repression of the target mRNA. Agomirs can also be used as enhancers of transcriptional repression or as miRNA replacement therapies [82].

### 2.7. Small Interfering RNAs (siRNAs)

Small interfering RNAs (siRNAs), also known as short interfering RNAs, are a class of non-coding double-stranded RNAs [83]. These molecules are typically 18–24 nucleotides in length, with a 3′ end that is hydroxylated and a 5′ end that is phosphorylated. The siRNA duplex also has a couple of overhang nucleotides. Classically, siRNAs are designed to consist of a characteristic 19+2mer structure, that is, a duplex of two 21-nucleotide-long strands with 19 complementary bases and a 2-nucleotide overhang at the 3′ end [84]. siRNAs are delivered to the cell in the form of a double-stranded RNA duplex, which includes a guide (antisense) strand and a passenger (sense) strand. The 3′ end of the passenger strand is often conjugated with peptide, lipid, or other inorganic moieties to increase cellular uptake levels of the siRNA [83,85,86,87,88]. The sequence of the guide strand is precisely complementary to the target sequence. The guide strand separates from the passenger strand once it reaches the cytoplasm and binds to endogenous Argonaute 2 (Ago2) protein, the essential nuclease of the RNA-induced silencing complex (RISC). The guide strand subsequently directs this protein complex to the target mRNA [89]. Unlike gapmers, which first bind to the target sequence before recruiting the nuclease, siRNA directly binds to the nuclease complex (RISC) and targets the mRNA. Due to their ability to degrade mRNA and silence specific gene expression, siRNAs have been widely used for knocking down protein expression and gene function analysis. Theoretically, synthetic siRNAs can silence any gene at any stage of life, making them highly potential candidates for therapeutic use.

siRNAs are typically composed of PS-backboned, sugar-modified nucleotides, e.g., 2′-OMe, 2′MOE, 2′-F, and LNA, among others [8,90,91,92]. Combinations of sugar-modified nucleotides are also in use [83,93]. Various enhancements have been made to the traditional siRNA design to improve silencing efficacy and target recognition and to reduce toxicity and immunogenicity [94]. These enhancements include the use of Dicer-substrate siRNAs, small internally segmented siRNAs, and asymmetric and hydrophobic self-delivering siRNAs [94,95,96]. However, the structural requirements of Ago2 and the RISC complex limit the extent to which siRNA can be chemically modified, which in turn limits the optimization of synthetic siRNA stability and cellular uptake levels through chemical modification. To date, five siRNA-based drugs have demonstrated clinical success and received FDA approval [97]. Patisiran, a modified siRNA with a lipid nanoparticle (LNP) carrier, was the first siRNA-based drug to receive FDA approval in 2018 [98]. It is indicated for the treatment of polyneuropathy in people with hereditary transthyretin-mediated amyloidosis (hATTR). Patisiran targets a sequence within the transthyretin (TTR) mRNA that is conserved across wild-type and all TTR variants to decrease the hepatic production of mutant and wild-type TTR. The other four siRNA-based drugs that received FDA approval are givosiran, lumasiran, inclisiran, and vutrisiran [99,100,101,102]. These drugs have been developed and marketed by Alnylam Pharmaceuticals (givosiran, lumasiran and vutrisiran) and Novartis (inclisiran) for different indications.

### 2.8. Choice of Antisense Oligonucleotide Platforms for Therapeutic Applications

As apparent from the above discussions, each ON platform has its own strengths and limitations. Choosing the optimal one requires careful consideration of various factors, e.g., ON chemistry, length, primary cellular/tissue target, administration routes, binding efficiency, uptake level, and potential adverse effects. Table 1 provides a simple comparison of different ON platforms.

ON chemistry determines the stability, binding affinity, and pharmacokinetic properties. ON length impacts the binding efficiency and specificity, dose determination, potential adverse effects, and the cost of an ON drug treatment. The primary cellular/tissue target is also crucial in determining ON efficacy and specificity. Some ON platforms perform better in targeting certain tissues/cell types, while others perform better in reaching other cells/tissues and getting taken up into these cells. In addition, administration routes and delivery strategies are critical in determining the efficacy and safety of ONs. 

There is no one-size-fits-all answer to the question of which ON platform is the best. Optimizing all these factors is essential to identify the most optimal ON platform for a specific therapeutic application. Various methods can be employed to improve the “optimumness” of the platforms for efficacy, delivery, and adverse effects, e.g., chemical modifications, bioconjugation with different moieties, and utilizing diverse delivery strategies.

## 3. Challenges Associated with the Therapeutic Use of ON Drugs

Despite substantial progress, ON drug therapeutic use faces several hurdles, particularly in systemic delivery (Figure 3). These challenges are associated with the physiochemical properties of ON molecules, including their high molecular weight and size, negative charge, and serum instability [109,110]. These properties render ONs prone to rapid nuclease degradation in biological fluids, triggering inflammatory and immune responses and leading to renal clearance and short blood circulation times [109,111,112]. Opsonization exacerbates these challenges by promoting rapid RES phagocytic cell uptake, resulting in higher ON concentrations in organs, such as the liver and spleen, complicating systemic ON delivery [113,114]. This uneven distribution contributes to the difficulties associated with systemic ON delivery [113,114].

Alternative routes, e.g., SC, IP, intrathecal, oral, or topical, also face significant challenges [13]. While they may offer advantages, such as reduced systemic exposure and improved tissue targeting, they still encounter obstacles, including ON physicochemical properties, cellular uptake, and endosomal entrapment. Moreover, capillary endothelial permeability and structure vary across organs and tissues, making some more accessible to ON therapeutics than others. The blood–brain barrier (BBB) poses a significant delivery barrier to the central nervous system (CNS) due to non-fenestrated capillaries, dense junctional proteins, and continuous basement membranes, making large molecule extravasation into the brain extremely challenging [112]. Cellular uptake is another ON drug delivery challenge, hindered by cell membrane and ON molecule physicochemical properties. As a result, the primary internalization mode is gymnosis (Figure 4). Once inside the cell, the ON drug must escape endosomal entrapment for release into the cytoplasm, presenting another obstacle to ON drug activity [109]. Developing effective strategies to overcome these challenges is vital for successful ON drug delivery through any route.

## 4. Gymnotic Uptake of ONs

Naked ONs typically do not easily permeate the hydrophobic cell membrane. However, through appropriate dosing, cells can internalize naked ONs through an endocytic process known as gymnosis (Figure 4) [115,116,117]. The mechanisms behind this process are not fully understood and can be classified as productive or unproductive based on whether it yields a functional outcome in recipient cells [118]. The uptake process can be sketchily divided into three stages: association, internalization, and trafficking [119,120].

During association, ONs contact cell-surface proteins. PS-modified ONs exhibit higher cell surface-binding potential compared to unmodified ONs. Scavenger proteins on endothelial cell membranes can bind to certain ON species [25,119]. Particularly, class A scavenger receptors (SCARAs) are principal association targets for peptide-conjugated PMOs, tcDNA, and 2′-OMe ONs. These findings were expanded to include SCARAs as binding targets for PS-ONs when administered in high concentrations [121,122,123,124]. Other protein receptors e.g., stabilin-1 and stabilin-2, have been identified as binding PS-ONs with high affinity, inducing clathrin-dependent endocytosis [125,126]. Additionally, the epidermal growth factor receptor (EGFR) has been shown to directly interact with PS-ONs, co-trafficking them alongside EGF into clathrin-coated pit structures [127]. Transmembrane SID1 proteins also contribute to ON gymnosis uptake, but their exact role is yet to be determined [128]. Protein binding is a crucial step in this process, while plasma membrane lipids coordinate various functions by laterally segregating membrane proteins into lipid raft structures.

Following association, naked ONs are internalized upon binding to a cell surface protein receptor. After binding, the ON enters the cell through endocytosis, either clathrin-dependent or caveolae-mediated. Clathrin-dependent endocytosis is the most common form, involving clathrin-coated pits [129,130]. In contrast, caveolae-mediated (clathrin-independent) endocytosis involves caveolae and is specific to certain cell types [131,132]. Along with these types, ONs can be internalized through micropinocytosis [16,120,131]. Micropinocytosis is a type of endocytosis that does not require association with a protein receptor on the cell surface. Instead, it involves the formation of a protrusion on the cell membrane that engulfs a large volume of the extracellular environment. This internalized volume is large enough for nonselected solute molecules to be taken into the cell. Macropinocytosis is an activated process; however, it internalizes membrane patches that are much larger than other endocytic routes [131].

After internalization, ONs are transported throughout the cell to reach their ultimate destination: the cytosol or nucleus. This trafficking process is orchestrated by a complex network of cytosolic and nuclear proteins directing ONs to specific targets [133]. ONs must escape endosomes to elicit their function. Endosomal escape is the process of extricating ONs from endosomal compartments. ONs enclosed within endosomes may get damaged during this process as endosomes are intrinsically acidic compartments.

The endolysosomal network has three potential outcomes: exocytosis, cytosol release, or lysosomal degradation [134]. The optimal ON pathway is to exploit the endosomal pathway by traversing toward the nucleus and escaping prior to reaching the lysosomes [135]. Prominent endosomal escape strategies involve acidifying endosomal vesicles as they mature and buffering the lumen of late endosomes and lysosomes with an ionizable or amphipathic delivery agent, known as the “proton sponge effect”, inducing osmotic inflow and endosome leakiness due to physical membrane stress [136]. Another strategy exploits endolysosomal organelle lipid profile differences, e.g., using LDL cholesterol to enlarge endosomes and increase their volume, potentially leading to leakage due to mechanical stress [137,138,139]. In addition, multivesicular body (MVB) formation and phospholipid lysobisphosphatidic acid (LBPA) play crucial roles in controlling intraluminal vesicle (ILV) fusion cycles and promoting productive ON release [140,141,142,143].

Endosome escape remains one of the largest physiological barriers to ON therapy on a pharmacological basis. Recent research focuses on devising strategies to manipulate endocytic uptake stages to facilitate ON endosomal compartment escape. Various endosomal escape-promoting strategies are under development, e.g., pH-sensitive peptide and polymer bioconjugation, small molecule enhancers, nucleotide chemical modification, and liposome and other carrier-based ON delivery systems [135,143,144,145,146]. These strategies aim to enhance ON delivery efficacy and specificity to target cells by enabling ONs to bypass degradation and reach their intended targets.

## 5. Chemical Modifications to Enhance Stability, Cellular Uptake, and Safety

Modifying the nucleic acid backbone, the ribose sugar, and the nucleobase itself has been extensively employed to improve the drug-like properties of ON drugs while keeping them sufficiently ‘native’-like so they can perform their biological function with which they are assigned [13,147]. Natural nucleotides face limitations, e.g., nuclease degradation, poor cellular uptake, and rapid blood clearance. Since the discovery of non-natural nucleotides in the 1960s, extensive efforts have been made to identify natural and synthetic compounds with improved persistence in the plasma, resistance to nucleases, increased binding affinity to targets, and improved pharmacokinetics, pharmacodynamics, and biodistribution properties. 13 ON drugs have so far received approval from the U.S. Food and Drug Administration (FDA), and among these, nine do not require a delivery strategy and instead rely on chemical modifications for tissue delivery, illustrating the effectiveness of chemical modifications in enhancing the clinical value of ON drugs.

### 5.1. Backbone Modification

The backbone of naturally occurring DNA or RNA molecules is composed of phosphate groups and sugars connected by PO linkages. However, these linkages are the target of degradative endo- and exonucleases. The nonmodified PO linkage has several disadvantages for therapeutic use, including a short half-life in circulation due to susceptibility to nuclease degradation and low serum protein-binding ability. To overcome these limitations, therapeutic ONs often incorporate phosphorothioate (PS) linkages (Figure 2), in which a sulfur atom replaces the non-bridging oxygens of the inter-nucleotide phosphate group [28]. Other types of backbone modifications, e.g., methylphosphonate and boranophosphate, are less common [148]. Interestingly, PS linkages are not entirely alien to nature [149]. Some bacteria in nature have CpG motifs that contain partially modified PS backbone linkages. The binding of CpG dinucleotides to TLR9 has been found to stimulate the immune system and activate B-cells [150]. This discovery has served as the basis for many therapeutic developments, including ISOs [151].

PS-modified ONs are well tolerated and do not interfere with RNase H activity; however, PS-modified siRNAs are less active than equivalent phosphodiester (PO) siRNAs [152]. Therefore, siRNAs typically contain PS modifications only at the termini, if at all [83,153]. Scavenger receptors, e.g., stabilins, aid in the cellular uptake of sulfated molecules. ONs with PS linkages or thiol tails can be internalized into organs where scavenger receptors are highly expressed, e.g., the liver [125,154]. PS linkages improve both nuclease resistance and protein binding in plasma and cells [155]. This modification also increases the interaction between ONs and plasma proteins, which helps increase the plasma circulation time and reduces renal clearance, thus improving pharmacokinetics. However, the binding of PS-modified gapmers to plasma α2-macroglobulin (A2M) is not as effective [156]. In addition, PS modification improves the interaction of ONs with intracellular proteins, e.g., nucleolins, potentially promoting ON accumulation in the nucleus [157,158] This resistance to cellular nucleases results in longer tissue retention of the ONs. The addition of one or more PO links can adjust ON longevity by reducing its nuclease stability in situations when this is desired, e.g., when extended gene silencing may be toxic [159]. The main disadvantage of PS modification is that it increases persistence at the cost of reduced ON-target binding. However, the addition of other modifications can help overcome this issue.

PMOs have a modified backbone in which synthetic, noncharged morpholine connections replace the PO linkage and the ribose sugar ring [160]. The benefits of this chemistry include high efficacy and specificity, nuclease resistance, water solubility, and reasonable production costs [161]. However, PMOs have reduced serum protein binding, leading to rapid blood clearance and limited tissue distribution. All four FDA-approved PMO drugs face this issue, as most PMOs are estimated to be eliminated from the kidney within hours of intravenous injection [48,49,162].

Controlling the stereopurity of ON backbones also offers potential improvements. The introduction of a PS-linkage creates a chiral center at each modified phosphorus atom (designated *S_p_* and *R_p_* isomers). As such, a fully PS-backboned ON of 20 nucleotides is a racemic mixture of the 2^19^ (=524,288) possible permutations. The physicochemical properties of each stereocenter are distinct in terms of hydrophobicity/ionic characteristics, nuclease resistance, target affinity, and RNase H activity. Stereopurity of the PS-backbone impacts the thermal and metabolic stability, lipophilicity, and biodistribution of steric blocking ONs and gapmers [162,163,164,165,166]. Stereopurity is also essential for aptamer activity, as some chiral architectures of nucleic acids are not substrates for endogenous nucleases (called spiegelmers) [167]. Therefore, aptamers must be synthesized with chiral structures recognizable by endogenous enzyme systems to exert therapeutic effects. Fortunately, SELEX can be engineered and optimized to identify specific aptamers with specific chirality only [168]. However, achieving stereopurity for other types of ONs, e.g., steric-blocking ONs and gapmers, is challenging from a manufacturing perspective. The stereoisomer-specific effect of ONs indicates that the racemic mixtures of ON drugs currently approved or under development contain only a small fraction of therapeutically optimized drug molecules, with the rest being stereoisomers with only modest or no activity. ON drugs would be advanced significantly by identifying the most effective stereoisomers, allowing for lower doses of more potent drug molecules.

### 5.2. Nucleobase Modification

For a long time, researchers have been exploring methods to modify the chemistry of nucleobases in order to achieve optimized Watson–Crick base-pairing and thereby control the melting temperature of the ON [147]. By altering the nucleobase chemistry, the ON affinity for its target is increased, resulting in a thermally more stable ON–target duplex. This thermal stability is crucial for splice-switching ONs, as the stronger the ON hybridizes with its target, the better it can mask the splice site or inhibit the assembly of the ribosomal complex, thereby preventing translation [169].

A frequent site for nucleobase modification is the 5-position of pyrimidines [170]. Substituting a single base with a methyl pyrimidine (e.g., 5-methylcytidine or 5-methyluridine/ribothymidine) can raise the melting temperature of an ON by approximately ~0.5 °C [147]. It is believed that stacking the methyl groups between the nucleobases in the major groove of the formed duplex is responsible for the increased melting temperature and stability. However, if the modified nucleobase is too large, it may negatively impact ON activity. For example, 5-propynyl bases, a typical 5′ position modification, may hinder siRNA-mediated silencing as they are bulky and can resist RISC from attaching precisely [171]. In addition to changing the nucleobases, abasic nucleotides (i.e., nucleotides without no nucleobases) have been employed for both the allele-specific suppression of mutant alleles and to disrupt miRNA-like silencing while preserving the on-target slicer function [172,173].

### 5.3. Sugar Modification

Modifying the deoxyribose sugar in DNA and ribose sugar in RNA can increase ON stability and improve pharmacokinetic properties [13]. The 2-carbon (2′C) position of the ribose sugar is frequently modified in synthetic ONs. The electron-withdrawing group at the 2′C position causes the ribose to adopt a conformation that is favorable for duplex formation, making RNA–RNA duplexes generally more stable than DNA–DNA duplexes [12]. As a result, most efforts to modify the sugar of ON aim to adopt this structure. A wide array of synthetic and natural sugar-modified ON chemistries have been identified over the past few decades [8,12,174]. However, hybridization analyses have shown that several 2′C modifications do not consistently increase ON stability and affinity for its target [175]. The compatibility of sugar-modified chemistries for therapeutic development fundamentally depends on the mechanism of action of a given ON platform, the design of the ON, and the expected biodistribution of the ON [8]. The most widely utilized sugar-modified ON chemistries include 2′-OMe, 2′MOE, and 2′-F, all created by substituting the hydroxyl group at 2′C in the ribose sugar (Figure 2). These modifications increase nuclease resistance by replacing the naturally occurring nucleophilic 2′-hydroxyl group of naturally occurring RNA. Substituting the 2′-hydroxyl group improves plasma stability, and internalization into cells, and increases tissue half-life to prolong the drug’s effects. These modifications also enhance the binding affinity of ONs for their targets by promoting the 3′-endo-pucker conformation of ribose [176,177].

Not all sugar modifications suit all ON types and mechanisms of action [8]. Substitutions at the 2′-carbon (2′C) position of the ribose sugar for steric blocking ONs and gapmers can increase binding affinity, but do not necessarily enhance delivery. However, for siRNAs, the ON must be recognized by the RNA interference (RNAi) machinery, remain within its loading capacity, and function properly, e.g., inactivating the passenger strand and promoting efficient hybridization between the guide strand and target mRNA [178]. Studies indicate that incorporating sugar-modified nucleotides at certain positions of the siRNA can provide significant stability but may impede RISC loading and RNA silencing, while using modified chemistry on the entire strand can improve biodistribution and effective mRNA silencing [178,179,180]. Thus, the use of modified chemistry in siRNA designs must be carefully optimized. Studies indicate that alternating 2′-F and 2′-OMe-modified nucleotides can significantly increase siRNA potency [181,182]. Additionally, Alnylam’s enhanced stability chemistry design, which incorporates a higher proportion of 2′-OMes and two PS linkages at the 3′-end of the guide strand and the 5′-end of the passenger strand, has been found to be highly effective [181,183].

Along with the modifications made to the 2′C residue, the sugars in nucleic acids may also be modified in such a way that “locks” the nucleotide into a specific conformation, known as a bridged nucleic acid (BNA). This bridge is typically incorporated synthetically using a methylene linkage between the 2′C and 4′C of the sugar, resulting in a constrained and rigid structure. Since the first report of BNA synthesis by Takeshi Imanishi’s group in the late 1990s, various variations of BNA have been synthesized and effectively used in various types of ONs, e.g., siRNAs, gapmers, splice-switching ONs, and antagomirs [184,185]. Common variations of BNA include locked nucleic acid (LNA), 2′,4′-constrained 2′-O-ethyl BNA (cEt), tricyclo-DNA (tcDNA), and 2′-O,4′-C-ethylene-bridged nucleic acids (ENAs) [186,187,188,189] (Figure 2). The bridged conformation of the nucleotides increases the melting temperature of the ONs significantly, e.g., a 2–8 °C increase per modified nucleotide depending on the length and class of ON. This helps improve both the nuclease resistance and ON affinity for the target [190]. The selection of BNAs for ON design primarily depends on the ON platform mechanism of action and desired biodistribution. For example, the 3-10-3 cET-DNA-cET gapmer design is more efficient than the 5-10-5 2′MOE-DNA-2′MOE design targeting the same mRNA region [191]. However, the non-natural structures of certain BNA variations, e.g., cET and tcDNA, have limitations in RNase H or RNAi applications, restricting their use in gapmers and siRNAs, while others, such as LNA, are highly effective for various ON types [185,192]. Furthermore, tcDNAs, a popular constrained DNA analog, show potential for crossing the blood–brain barrier (BBB) [193] and have been used in gapmer-flanking regions [194].

## 6. Bioconjugation

Bioconjugation, or the covalent linking of various molecules to ONs, is a strategy for improving the targeting of tissues and enhancing the uptake of ONs at the cellular level [195]. Bioconjugates are single-component entities with precise stoichiometry and well-determined pharmacokinetic properties [11,103,195]. Molecules commonly used for bioconjugation to ONs include lipids (e.g., cholesterol for better interaction with lipoproteins in circulation), peptides (e.g., cell-penetrating or cell-specific targeting peptides), aptamers, antibodies, and sugars (e.g., GalNAc) [196,197,198,199,200,201,202,203]. They play crucial roles in cell and tissue recognition, cellular internalization, and pharmacokinetics of the ON [195]. Although the mechanism by which bioconjugation improves ON activity is not fully understood, studies suggest increased cellular uptake and endosomal destabilization, leading to sustained therapeutic outcomes [11,195,204,205,206]. ON bioconjugates, synthesized using chemical methods (e.g., click chemistry) without requiring extensive characterization analyses [207,208,209,210], improve the biodistribution of ONs compared to that with free ONs [203,207,208,209]. Bioconjugates have also been shown to alter the kinetics of therapeutic ONs [202,204]. As a result, bioconjugates are increasingly used in ON therapeutics. Four out of the five siRNA drugs approved so far by the FDA, contain bioconjugated molecules.

### 6.1. Peptide Conjugates

Peptides offer an appealing source of ligands for therapeutic ON bioconjugation, imparting tissue/cell-specific targeting, cell-penetrating, or endosomolytic properties [206,211]. Cell-penetrating peptides (CPPs) can transport various cargoes across cell membranes and biological barriers [206,212]. CPPs, composed of 4 to 40 amino acids, are often referred to as protein transduction domains [206,213]. Classified based on their physicochemical properties into cationic, amphipathic, and hydrophobic peptides [213,214], CPPs have two main mechanisms of action: receptor-mediated endocytosis and direct translocation [106,214]. Receptor-mediated endocytosis is the most widely observed mechanism, where CPPs bind to specific cell surface receptors, triggering endocytosis [215,216]. This mechanism allows CPPs to specifically target certain cells or tissues. Direct translocation does not rely on binding to specific receptors; instead, CPPs use the cell’s energy to form transient pores in the cell membrane, allowing the CPP to cross and deliver their cargoes directly to the cytoplasm or nucleus [145,217,218,219].

One promising application of CPPs is their ability to conjugate directly with charge-neutral ON chemistries e.g., PMO and PNA [106,206,213,214]. These conjugates have shown efficacy in targeting viral and bacterial infections, as well as specific tissues, such as the skeletal muscle, heart, and central nervous system (CNS) [220,221,222]. Exon-skipping PMOs conjugated with cationic peptides, e.g., (RXR)4-PMO and the ‘B’ peptide (RXRRBR)2XB, have demonstrated efficacy in treating DMD [223,224]. Recent studies have shown that exon-skipping PMOs conjugated to a newly developed CPP, DG9, are highly efficient in bypassing the blood–brain barrier and targeting cardiac muscles in DMD and spinal muscular atrophy (SMA) disease models [225,226]. Researchers have also created chimeric peptides that combine muscle-targeting peptides with existing CPPs, increasing the efficacy of the treatment [227]. Additionally, new PMO/PNA internalization peptides (Pips) have been developed, demonstrating potency in animal models, and reaching critical tissues, such as the cardiac muscle, in DMD [196,228,229,230].

### 6.2. Lipid-Based Conjugates

Lipid-based moieties, e.g., cholesterol and its derivatives, have been studied for enhancing the delivery of siRNAs and antagomirs [103,231]. Cholesterol bioconjugation improves in vitro delivery by promoting endosomal release, prolonging plasma half-life, and increasing liver accumulation upon systemic administration [232,233,234]. Cholesterol-conjugated siRNAs have been used for silencing genes in the liver, e.g., Apolipoprotein B (ApoB) [235], and more recently, in targeting myostatin (*Mstn*) in skeletal muscles, an organ that has historically been difficult to target [236,237]. Other lipid derivatives, e.g., α-tocopherol (vitamin E) and long-chain (>C18) fatty acids via a trans-4-hydroxyprolinol linker, have also been explored for enhancing siRNA delivery [233,238].

The in vivo activity of lipid-conjugated siRNAs depends on their ability to bind lipoprotein particles, e.g., high-density lipoprotein (HDL) and low-density lipoprotein (LDL), thereby leveraging the body’s endogenous lipid transport and uptake system [238,239]. Pre-assembling cholesterol siRNAs with purified HDL particles enhance gene silencing in the liver and jejunum [238]. Lipoprotein particle pre-assembly also affects siRNA distribution, with LDL siRNA particles primarily taken up by the liver and HDL siRNA particles by the liver, adrenal glands, ovary, kidney, and small intestine [238]. The endocytosis of cholesterol siRNAs is mediated by scavenger receptor type B1 (*SCARB1*, *SR-B1*) or LDL receptor (*LDLR*) for HDL and LDL particles, respectively [238]. Hydrophobicity affects in vivo siRNA association with lipoprotein classes, with more hydrophobic conjugates binding preferentially to LDL and less hydrophobic conjugates to HDL [233].

### 6.3. Receptor–Ligand Conjugates

Receptor–ligand conjugates facilitate specific binding to target cell receptors and mediate tissue-specific ON delivery. The trimeric GalNAc is the most clinically successful tissue-targeting ligand used in ON development to date [183,240]. GalNAc bioconjugations have recently emerged as a breakthrough in liver-targeted nucleic acid therapeutics. The asialoglycoprotein receptor (ASGR), present on the sinusoidal surface of liver cells, is a high-capacity, rapidly internalizing receptor specifically and abundantly expressed on hepatocyte surfaces [240,241]. ASGR facilitates the clearance of desialylated serum glycoproteins containing a terminal galactose or GalNAc [242,243]. GalNAc conjugates bind to the ASGR receptor in the presence of calcium ions and at pH > 6, resulting in the internalization of conjugates via clathrin-dependent receptor-mediated endocytosis [241,244]. Once inside the cell, the GalNAc conjugates dissociate from the ASGR and are degraded in the lysosome, while the ASGR recycles back to the cell surface [241,244,244]. This enables targeted ON delivery in the liver, and the heavy blood flow and fenestrated endothelium of the liver support prolonged efficacy with a single injection [241,245].

Significant improvements have been achieved with clinically relevant modifications of ONs, e.g., GalNAc-siRNA and GalNAc-ON conjugates [245]. Various chemical modifications, such 2′-F or 2′-OMe nucleotides, have been explored to improve drug-like properties. Studies have shown that fully modified siRNAs exhibit higher potency and prolonged duration of gene silencing in vivo and in vitro. However, partially modified siRNAs (fewer than 70% of the total nucleotides) have also been used extensively to evaluate the efficacy of various bioconjugates based on siRNA distribution and in vivo efficacy. A reduction of 2′-F content (less than 20%) yielded excellent in vitro activity and in vivo performance of GalNAc-siRNA [240,241]. The incorporation of metabolically stable 5′-(E)-vinylphosphonate into the 5′ end of the antisense strand of siRNA resulted in significantly improved in vitro potency and stability [240]. Additionally, incorporating glycol nucleic acid (GNA) in the antisense strand seed region of GalNAc-siRNA reduced off-target toxicity while maintaining on-target activity and increased thermal and metabolic stability [240].

The proximity of GalNAc is crucial for effective ASGR recognition [245]. Clustering GalNAc and locating it near the 3′ end of the siRNA enhances the binding affinity and potency of the conjugate [201]. A more recent study suggested that a higher number of GalNAc sugars, e.g., trivalent, and tetravalent assemblies, show more potent gene-silencing effects in the liver than bi-, tri-, and tetravalent linear assemblies of GalNAc [246]. This highlights the importance of the number and positioning of GalNAc sugars for conjugate efficacy. Similarly, studies have shown that the valency and position of the GalNAc conjugate are important for the delivery efficacy and potency of GalNAc-linked ON drugs. The introduction of a tri-antennary GalNAc conjugate enhances the binding of ONs to the ASGR and improves their potency [247]. The position of the GalNAc conjugate near the 3′ end of the ON enhances the binding affinity and potency of the conjugate [245,248,249].

Overall, the use of GalNAc conjugates as a targeted delivery strategy for ONs in the liver has the potential to revolutionize the treatment of liver diseases. This approach allows for efficient uptake and prolonged efficacy with a single injection, and the various chemical modifications that have been explored have further improved the properties of these conjugates [250,251,252]. However, further research is needed to fully understand the mechanism of action and enhance the safety and efficacy of GalNAc-conjugated therapeutics.

### 6.4. Antibody and Aptamer Conjugates

Antibody–RNA bioconjugates offer a promising strategy for nucleic acid therapeutics by attaching monoclonal antibodies or antibody fragments to functional ONs, e.g., siRNA or miRNA [25,253]. They have been used for imaging and protein detection and hold potential for therapeutic ON delivery due to their tissue-specific targeting ability. Aptamers offer similar opportunities for targeted ON delivery [254,255].

Antibody–RNA bioconjugates are still in early development [256], but interactions between antibodies and specific cell surface receptors have the potential to facilitate ON delivery to otherwise inaccessible tissues and cell subpopulations. Various receptors have been targeted for siRNA delivery, e.g., HIV gp160 protein, HER2, CD7, CD71, CD44, and TMEFF2 [203,257,258]. Steric-blocking ONs have also been conjugated with antibodies against CD44, EPHA2, and EGFR [259]. Aptamers have shown similar promise in mediating the delivery of therapeutic ONs as aptamer–ON conjugates or within nanoparticle formulations [260,261]. Aptamer-ONs have since been shown to be effective for the in vivo delivery of miRNAs, antagomirs, steric-blocking ONs, and bi-modular miRNA–antagomirs within preclinical cancer models. Apoptosis-inducing siRNA bioconjugated with aptamers designed to target prostate-specific membrane antigen-expressing cancer cells have recently shown potential. Current research on aptamer-ONs involves chemical modifications to protect the ONs from nuclease degradation and increase their plasma half-life.

Avidity Biosciences and Dyne Therapeutics are currently developing antibody and aptamer bioconjugates for skeletal and cardiac muscle targeting [262,263,264].

### 6.5. Polymer Conjugates

PEG is widely used to prolong the blood circulation time and improve drug efficacy [265]. PEG is a highly versatile, non-ionic, and hydrophilic polymer that can be functionalized at its end groups. Drugs conjugated with PEG have been shown to demonstrate improved pharmacokinetic and pharmacodynamic properties in terms of the chemical absorption, distribution, metabolism, excretion, and toxicity (ADMET) features of the drug [265]. PEGylation, the process of covalently attaching PEG to a drug, has been utilized for therapeutic proteins and ONs, e.g., pegaptanib [265,266]. PEGylation results in a protective hydration shell around ONs, increasing their stability and reducing renal excretion [265,267]. Recently, alternative polymers to PEG have gained attention for enhancing ADMET properties and immunogenicity, including poly(glycerol), poly(2-oxazoline), poly(amino acid), and poly[N-(2-hydroxypropyl)methacrylamide] [267,268].

The ADMET properties of PEGylated ONs depend on the PEG moiety’s physicochemical properties, including molecular weight, end-group modification, and architecture [265,269]. Pegaptanib contains a 40 kDa Y-shaped PEG, which decreases aptamer binding affinity four-fold but increases antiangiogenic efficacy due to prolonged tissue residence time [266]. Moreover, the PEG layer around the ON can interfere with cellular absorption efficiency and biodegradability, so various de-PEGylation systems have been investigated [267]. The judicious selection of the appropriate PEG moiety for bioconjugation to ON drugs is essential for achieving the desired pharmacological effect [265,267,270].

### 6.6. The Optimal Bioconjugation Strategy

ON bioconjugation strategies aim to enhance cellular uptake, stability, pharmacokinetics, cell/tissue targeting, and binding efficiency for optimal efficacy and safety (Figure 5). However, no single strategy can achieve the optimal level of these desired effects.

Peptide conjugates enhance cellular uptake and pharmacokinetics but may have limited targeting specificity. Lipid-based conjugates improve stability and cellular uptake but may induce toxicity or immunogenicity. Receptor–ligand conjugates enable targeted delivery but can be limited by cell surface marker variability. Antibody and aptamer conjugates enable targeted delivery but may have restricted tissue penetration and biodistribution due to their large size. Polymer conjugates can improve stability and cellular uptake but may also induce toxicity or immunogenicity.

The choice of bioconjugation strategy depends on application goals, target tissue and molecular target, pharmacokinetic profile, and safety considerations. Combining and utilizing these strategies in an integrated approach may achieve optimal outcomes surpassing those achievable with any single strategy alone. Further research is necessary to optimize the design and development of bioconjugates for ON therapeutics and to overcome the challenges of delivering these therapeutics effectively and ensuring their efficacy in clinical settings.

## 7. ON Delivery Systems

Carrier-based delivery systems have become an important tool in the field of ON-based therapeutics, offering numerous benefits over traditional delivery methods [271]. The pharmacological properties of these systems are largely dependent on the properties of the delivery system itself [272]. Desired properties can be built into the system through formulation design, resulting in multifunctional advanced drug delivery systems. One key benefit is their ability to protect ON cargo from premature degradation, increasing the effect duration and enhancing targeting [272]. Targeting can occur passively or actively. Passive targeting utilizes the microanatomical features of tissues, while active targeting involves decorating the delivery system with active targeting ligands. Another benefit is their ability to facilitate intracellular delivery by enhancing cellular uptake, intracellular trafficking, and endosomal escape [271,272]. Various nanocarrier types have been investigated, but the current focus is on lipid-, polymer-, and peptide-based delivery systems and hybrids, as they have shown great potential in delivering ONs to targets of interest [273,274,275,276,277]. The complexity of these systems presents new challenges, including cost, manufacturability, safety, quality assurance, and quality control.

### 7.1. Lipoplex, Liposomes, and Lipid Nanoparticles

Lipid-based systems, including lipoplex, liposomes, and solid lipid nanoparticles, are promising colloidal nanocarriers for bioactive organic molecules, e.g., ONs [278,279]. These systems are known for excellent drug delivery capabilities, attributed to the protective outer layer of lipids that surround the ON cargo [272,278]. Lipoplexes are formed by the direct electrostatic attraction between polyanionic ONs and synthetic cationic lipids [280]. Despite their unstable nature, lipoplex formulations have been effectively utilized for localized delivery applications. Liposomes and LNPs, while similar in design, differ slightly in composition and function [272,278]. LNPs for the delivery of ONs are typically composed of ionizable cationic lipids, phospholipids, PEG, and cholesterol derivatives [279,281]. LNPs, perhaps the most advanced in terms of clinical translatability for ON delivery, have been demonstrated to possess a high level of efficiency for ON drug delivery and the ability to target specific tissues [272,278,281]. LNPs are able to protect ONs from degradation and increase their circulation half-life, making them a promising option for treating various diseases [278]. Additionally, LNPs have shown a low level of toxicity, further highlighting their potential as a drug delivery system [278,281].

The recent approval of patisiran, an siRNA directed against transthyretin mRNA in an LNP system, has renewed interest in LNPs [11,278,279,282,283]. LNPs form polyelectrolyte complexes by facilitating ionic interactions between the positively charged functional groups and the negatively charged phosphate fraction of the ONs [284]. ONs can be encapsulated within the matrix of the LNPs or attached to their surface through covalent or ionic bonding [285,286]. Ionizable lipids are being developed for in vivo applications to avoid toxicity [287]. PEG lipids are often prudently added to the LNPs to stabilize the structure during manufacturing and storage [279,288]. Adding PEG lipids to LNPs increases circulation half-life, essential for effective in vivo delivery [288]. However, PEG can inhibit cellular transfection, so PEG lipids are designed to rapidly diffuse from LNPs after administration, allowing ON drugs to reach target cells without interference [288]. LNPs primarily accumulate in the liver, targeting liver cells responsible for targeted protein synthesis through apolipoprotein E-mediated surface adsorption via low-density lipoprotein receptors [288,289]. This allows the LNPs to specifically target liver cells responsible for the synthesis of the targeted protein [288,290]. After cellular uptake, endosomal escape of siRNA into the cytosol is facilitated through interactions between ionizable cationic lipids and anionic endogenous lipids in the endosomal membrane [291], enabling siRNA to silence the targeted gene.

### 7.2. Peptide-Based Delivery Systems

CPPs are a promising class of carrier-based drug delivery systems due to their ability to traverse cell membranes and are useful as drug delivery systems and bioconjugation ligands [25]. They can transport diverse cargoes, including ONs [25,206,292]. CPP-ON nanoparticle formation relies on synergistic interactions between cationic CPPs and anionic ONs [292]. Peptide-based vectors are more amphipathic and usually carry additional chemical modifications for ON encapsulation [206,292,293].

Common CPP modifications include hydrophobic additions, such as fatty acid derivatives [206,292,294], increasing formulation stability and enhancing cellular uptake and endosomal escape, crucial for ON effectiveness [295,296,297]. Various types of CPPs have been found to exhibit potential for ON delivery in a nanoparticle-based format, e.g., MPG and PepFect peptide derivatives [298]. Various CPPs, e.g., MPG and PepFect peptide derivatives, show potential for ON delivery in nanoparticle-based formats. Overall, CPPs provide efficient, targeted delivery and low toxicity.

### 7.3. Polymer-Based Delivery Systems

Polymer-based systems are attractive ON delivery carriers due to chemical flexibility, high structural integrity, and stability [108,299]. Synthetic polymers, such as polyphosphazenes, exhibit biocompatibility and chemical flexibility [300,301], can respond to external stimuli, enabling targeted ON release, and can be tailored to specific needs and delivery targets [108,302].

Poly(lactic-co-glycolic acid) (PLGA) is widely used for ON delivery, with high biocompatibility and the ability to form solid nanoparticles [303,304]. However, its anionic nature requires overcoming limitations through positively charged side chains or complexing with a positively charged moiety, such as polyethylene imine (PEI) [108,305], allowing stable, positively charged nanoparticles for ON encapsulation and protection. Natural biopolymers, such as chitosan, have also been used for ON encapsulation [306,307]. Chitosan, a cationic polysaccharide derived from crustacean exoskeletons, is often combined with anionic polymers, such as PLGA or alginate, forming complex nanoparticles [108,308,309], which can be engineered for specific properties, e.g., pH-responsiveness, allowing for targeted ON release [108]. Lipid-polymer hybrid nanoparticles have recently gained interest [108,310], combining serum stability, biocompatibility, and ON-loading capacity. They encapsulate ONs in their hydrophobic core, enabling targeted delivery to specific cells and tissues [310], with improved penetration through cell membranes, enhancing ON delivery efficiency.

### 7.4. Antibody Complexation Delivery Systems

Antibodies, widely studied for targeted therapeutics, offer natural specificity for certain cells or tissue types [264], increasing ON delivery specificity. One application is using antibody–avidin or antibody–protamine conjugates to deliver biotinylated ONs to specific targets [311,312]. The natural avidin–biotin complexation system allows for the binding of biotinylated ONs to antibody–avidin fusion molecules, effectively targeting therapeutic ONs [313]. The peptide protamine condenses siRNA, forming antibody–siRNA complexes [257,314]. This method links cytotoxic siRNAs with Her2-positive breast cancer cell-targeted antibodies, showing potential in cancer therapy [257,315]. Antibodies also protect therapeutic ONs from degradation, increasing stability and efficacy, making them appealing carrier delivery systems for ONs.

### 7.5. Extracellular Vesicle-Mediated Delivery Systems

Extracellular vehicles (EVs), including exosomes, microvesicles, and apoptotic bodies, are nanosized particles released by cells and play roles in cell communication, the immune response, and tissue repair [316,317]. They carry various molecules, such as proteins and nucleic acids, and have been explored as diagnostic and prognostic markers [316,318,319]. EVs have recently gained attention as potential ON drug delivery tools due to their non-toxic nature and potential for autologous production [320,321,322].

Exosomes, a subset of EVs, have numerous favorable properties for ON drug delivery [323,324]. They can traverse biological membranes, have longer circulation times, and are non-toxic [325,326,327]. Furthermore, some exosomes have inherent pro-regeneration and anti-inflammatory properties that may augment the effects of therapeutic ON delivery. However, a major challenge for exosome therapeutics is the efficient loading of therapeutic ON cargo. Endogenous loading methods, e.g., overexpression of the cargo in the producer cell line and exogenous loading methods, such as electroporation, sonication, and co-incubation with cholesterol-conjugated siRNAs, have been used [327,328,329,330,331,332,333]. The biodistribution of exosomes can be altered through surface ligands to target specific tissues [328,334,335,336].

While LNP-based delivery systems have already reached the clinic, EVs or exosomes to be more specific, are not far behind. The EV-based drug delivery field has grown exponentially in recent years, and the therapeutic applications of the engineered exosome-based delivery of ONs are presently at advanced preclinical and clinical trial stages [324,337].

### 7.6. Spherical Nucleic Acids

Spherical nucleic acids (SNAs) have gained attention as therapeutic materials due to their unique structure, consisting of a hydrophobic nanoparticle core and a hydrophilic nucleic acid shell [338,339]. The SNA structure typically consists of two components: a hydrophobic nanoparticle core and a hydrophilic nucleic acid shell [340]. The nanoparticle core of the SNA is composed of gold, silica, or other materials, and serves two purposes: (i) it modulates the SNA conjugate with unique physical and chemical properties, and (ii) it acts as a scaffold for the assembly and proper orientation of the nucleic acids [341,342]. The nucleic acid shell, on the other hand, is made up of short, synthetic ONs terminated with a functional group that can be utilized to attach them to the nanoparticle core [338,339]. The dense loading of nucleic acids on the particle surface results in a characteristic radial orientation around the nanoparticle core, which minimizes repulsion between the negatively charged ONs [339]. This nucleic acid shell is also responsible for imparting the chemical and biological recognition abilities of the SNA, including the therapeutic effects of the ONs [343,344]. This unique three-dimensional architecture formed by the nanoparticle core and the nucleic acid shell synergistically contributes to the chemical, biological, and physical properties of SNAs.

SNAs can be efficiently taken up by cells without positively charged co-carriers and have shown potential for crossing the blood–brain barrier (BBB) [345,346]. They have also been used for topical delivery in diabetic wound healing and psoriasis [347,348]. However, a major limitation that comes with ON delivery in the form of an SNA is that most SNA particles are deposited in the liver and kidneys, which may limit their application for certain diseases [339,349]. Additionally, the commercialization of SNAs is still in its early stages. Exicure, Inc. is currently commercializing SNAs as an ON-delivery strategy [350]. Exicure’s focus on developing SNAs for a range of disease indications and their ability to cross the BBB make them an attractive alternative to traditional nanoparticle-based delivery strategies. As research in this field continues to advance, it is likely that we will see more applications for SNAs in the near future.

### 7.7. DNA Nanostructures

DNA nanostructures, e.g., DNA origami, are utilized for ON delivery due to their self-assembly and precise geometries for fine-tuning properties [351]. Composed of long DNA molecules held in defined shapes by short DNA staples, they can be engineered to be small, enabling extrahepatic delivery [352,353]. Pristine DNA nanostructures preferentially accumulate in organs, such as the liver, kidneys, and lymph nodes, but can be modified for extrahepatic delivery [353,354]. Recent progress in chemical modifications of DNA nanostructures has expanded their potential for biosensing, bioimaging, and drug delivery [351,352,353]. These modifications create structures capable of efficiently delivering chemotherapeutic drugs and ON drugs, e.g., steric blocking ONs, siRNAs, and CpG ONs displayed on the structure surface [355,356]. However, future research must address challenges in optimizing the efficiency and safety of chemically modified DNA nanostructures [351,352,353]. Despite the excellent addressability and biocompatibility of DNA nanostructures, further research is needed to optimize their efficiency and safety.

### 7.8. Stimuli-Responsive Delivery Technologies

Stimuli-responsive nanotechnology offers innovative targeted ON delivery in response to various stimuli, e.g., pH, temperature, redox state, enzymatic activity, magnetic fields, and light [357,358]. This technology offers numerous benefits, including targeted delivery to specific sites, increased efficacy, and reduced off-target effects [357,358].

One promising approach is stimuli-activatable CPP conjugates [357,359] where a hairpin-folded peptide is covalently linked to an ON. Half of the hairpin peptide is positively charged, while the rest is negatively charged, enabling the CPP-ON conjugate to bind to cell membranes and neutralize itself until it reaches the target site where the conjugate is activated enzymatically [359]. This technology has been shown to be effective in the targeted delivery of siRNA to hepatoma cells both in vitro and in vivo [359]. Another notable approach is dynamic polyconjugates [357,360], which consist of ONs conjugated to a scaffold linked to delivery-assisting moieties. The acidic endosomal environment induces cleavage of the moieties, leading to endosomal escape. This technology has been shown to induce potent in vivo gene silencing after systemic injection [360,361].

DNA origami allows for the creation of sophisticated smart delivery vehicles for ONs [358], offering precise geometry and functionality. Aptamer-encoded nanobots interact with specific target proteins, releasing the ON payload [362]. These nanobots precisely target specific cells and tissues by engineering aptamers to bind to cell surface receptors or proteins [358,362]. This is achieved by engineering the aptamers to bind to specific cell surface receptors or proteins, allowing for highly selective uptake by the target cells. Additionally, the ability to control the geometry and functionality of the DNA nanostructures allows for the concentration of therapeutic payloads at the desired sites of action, leading to higher efficacy and reduced off-target effects [358]. Recently, there have been several studies demonstrating the potential of DNA origami for ON drug delivery, especially to target cancer cells in vivo, resulting in improved therapeutic efficacy [358,361,363,364].

### 7.9. Multilayered Biopolymer Nanopolymer-Based Carriers

Multilayered biopolymer nanopolymers have emerged as a promising strategy for localized nucleic acid-based therapy delivery [365,366]. The layer-by-layer assembly of thin multilayered films, e.g., polyelectrolyte multilayers (PEMs), has gained attention for immobilizing and releasing nucleic acids [367]. The layer-by-layer adsorption of oppositely charged polyelectrolytes on surfaces provides nanometer-scale control over incorporating DNA, RNA, and ONs into thin polyelectrolyte films [368]. This approach utilizes electrostatic forces between charged polymers and oppositely charged surfaces for film growth, achieved through stepwise exposure to polycation and polyanion solutions [368,369].

Incorporating biomacromolecules, such as DNA and proteins, into multilayered polyelectrolyte films offers a platform for macromolecular therapeutic release [368]. However, the large size of these biological polyelectrolytes hinders diffusion, necessitating encapsulation or direct incorporation into film structures [368].

Over the past decades, layer-by-layer techniques have been applied to fabricate multilayered films using DNA and nucleic acid constructs [368,369]. Recently, the focus has shifted to incorporating and releasing plasmid DNA constructs and functional ONs for therapeutic applications. DNA-containing multilayered films must allow for disassembly under physiologically relevant conditions [368]. Strategies to disassemble films or destabilize capsules include environmental variable changes, the incorporation of cleavable polyelectrolytes, receptor–ligand interaction-based films, and external stimuli application [367,368,370]. These methods offer spatial and temporal control over nucleic acid-based therapeutic delivery and hold promise for future biomedical applications. Further research is required to optimize this technology for ON delivery. However, research is required to optimize this technology for ON delivery.

### 7.10. Advantages and Limitations of ON Delivery Systems

Delivery systems for ONs have emerged as a promising approach to enhance the efficacy and safety of ON-based therapeutics. However, each delivery system comes with its own set of advantages and limitations (Table 2). Choosing an appropriate delivery system involves assessing efficiency, specificity, safety, and scalability, with factors including the ON type, target cell or tissue, desired therapeutic outcome, and safety considerations.

Researchers continually refine ON delivery systems to improve efficiency and safety. When combined with other enhancement strategies, such as chemical modifications or bioconjugation, delivery systems can significantly enhance therapeutic efficacy and minimize the potential adverse effects of ON drugs. Delivery systems should be regarded as a critical component of ON-based therapeutics to optimize therapeutic potential and minimize side effects. Ongoing research in this field is expected to lead to advancements in ON delivery systems and the development of innovative ON-based therapeutics with improved clinical outcomes.

## 8. Challenges, Considerations, and Future Perspectives

ON-based therapeutics show great promise in treating numerous disorders, targeting DNA, RNA, mRNA, and proteins. Recent regulatory approvals highlight the field’s maturity, but challenges remain, including delivery to target tissues, requiring more research and development. The combination of nucleic acid chemical modifications, conjugation to cell/tissue-targeting ligands, and nanoparticle carrier systems has enhanced the efficiency of ON drug delivery and enabled therapeutic molecules to reach previously inaccessible target tissues. The field is rapidly expanding, with only a small number of combinations of ON chemistries, targets, and formulations investigated to date, indicating that the field is still in its early days. However, ONs have already been proven effective in targeting DNA, RNA, mRNA, and proteins, firmly establishing them as a therapeutic class. The main hurdle for the application to a wider range of disorders is delivery to target tissues, which requires further research and development.

A significant concern is the cost associated with developing and using ON therapeutics. Nusinersen, for example, currently costs $750,000 for the first year and $375,000 in subsequent years [53]. Additionally, the annual cost for eteplirsen and three other similar PMO drugs, including golodirsen, viltolarsen, and casimersen, is $300,000 [45,48,49,52]. The cost–benefit ratio for highly effective, life-changing medications e.g., nusinersen, is likely to be favorable [371]. However, justifying the cost-effectiveness of drugs with limited efficacy, e.g., eteplirsen or golodirsen, presents a significant challenge [49]. To address the high costs, researchers could focus on targeted delivery strategies, potentially lowering required drug doses and costs. Novel delivery technologies, such as liposomes and nanoparticles, could enhance efficacy and reduce the cost–benefit ratio. Additionally, exploring cost-effective manufacturing methods, e.g., synthetic biology techniques and 3D printing, may help reduce costs [372].

Safety is crucial when considering ON therapeutics. The immunogenicity of delivery agent components or ligand conjugates poses challenges for safe and effective ON drug delivery. Thorough evaluations of each drug’s safety profile and delivery system are essential to meet regulatory requirements and minimize adverse patient effects. Long-term safety research and the development of sophisticated animal models and optimized in vitro test systems can improve our understanding of the safety and efficacy of these therapeutics and facilitate translation from the laboratory to the clinic.

## 9. Conclusions

The ON therapeutics field is experiencing significant growth, with increasing regulatory approvals for ON-based therapies. This growth results from optimizing nucleic acid chemical modifications, conjugation to targeting ligands, and developing delivery systems that have improved the efficiency and specificity of ON drug delivery. Despite progress, there is much to learn and develop in this field, with the limited number of investigated ON chemistries, targets, and formulations representing only a fraction of the potential for ON-based therapies. Further research and development are necessary to fully realize this potential. The challenge of delivering these therapeutics to target tissues remains a significant obstacle. Additionally, safety considerations and regulatory requirements for synthetic ON chemistries must be addressed before widespread clinical adoption can be achieved.

ON therapeutics offer a promising solution for a wide range of disorders, and their efficacy has been demonstrated across various targets, including DNA, RNA, pre-mRNA, and proteins. However, the field is still in its early stages. Continued research and development are necessary to optimize the efficacy and safety of ON-based therapies.

## Figures and Tables

**Figure 1 pharmaceutics-15-01130-f001:**
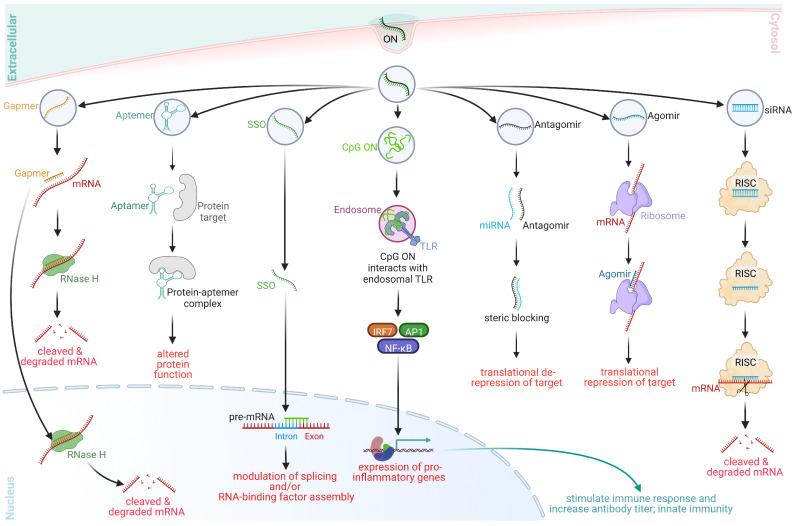
**ON-mediated gene regulatory mechanisms and their location of action.** The simplified mechanistic actions of (from left to right) gapmers, aptamers, steric blocking antisense ONs (SSOs), immunostimulant CpG oligodeoxynucleotides (CpG ON or CpG ODN), antagomirs, agomirs, and siRNA are shown with their respective cellular localization of activity. Gapmers induce the cytosolic or nuclear degradation of target mRNA, while aptamers modulate target protein function similar to monoclonal antibodies. SSOs modulate splicing and the assembly of RNA-binding factors, while CpG ONs stimulate the immune response. Antagomirs and agomirs, on the other hand, target miRNAs and their translational regulation, respectively. siRNA, on the other hand, triggers the RNA-induced silencing complex-mediated cleavage of mRNA (Created with BioRender: OK25720E3N).

**Figure 3 pharmaceutics-15-01130-f003:**
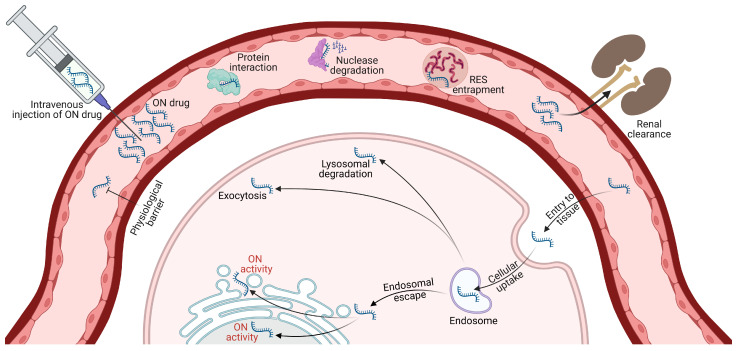
Challenges of systemic delivery of ON drugs. Most systemically administered naked ON drugs do not reach the intended tissue/cells. A small drug fraction evades nuclease degradation, RES phagocyte uptake, and renal excretion. Physiological barriers, e.g., the BBB, impede ON drugs from reaching their target. Once reaching the target tissue/cell, it is typically internalized through gymnosis. Following cellular uptake, ON drug molecules require endosomal entrapment escape to reach the target genetic material. Only a minute fraction of the originally administered ON drug effectively reaches its final target and exhibits its activity (Created with BioRender: GP257207MX).

**Figure 4 pharmaceutics-15-01130-f004:**
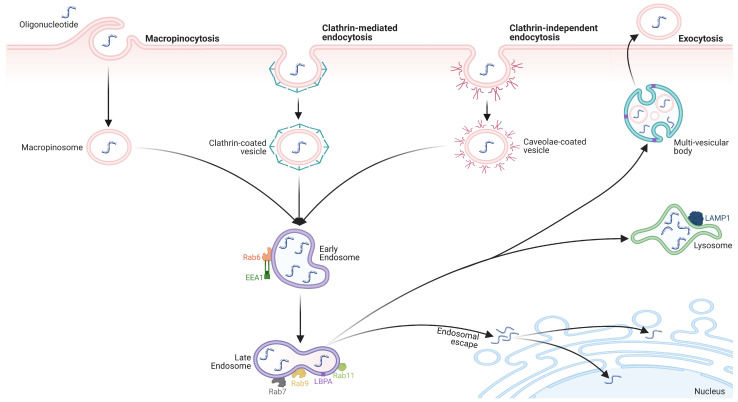
A simplified overview of the gymnotic ON uptake mechanism. ON uptake through gymnosis and subsequent endosomal escape is a complex and multifactorial process. Major identified ON internalization routes include clathrin-dependent endocytosis, clathrin-independent endocytosis, and macropinocytosis, each representing a distinct ON uptake mechanism into cells. Once internalized, ONs are transported through a series of endosomal compartments sequentially, first to the early endosome (pH range: 6–7) and then to the late endosome (pH range: 5.5–6). At this point, ONs may be directed towards the lysosome (pH range: 4.5–5) for degradation or towards multivesicular bodies for exocytosis. The late endosome membrane remodeling and transition to multivesicular bodies or lysosomes have been identified as potential points of endosomal escape, as these stages in the endosomal pathway are thought to represent opportunities for ONs to bypass degradation and reach their intended targets. Various markers have been proposed for tracking the endosomal pathway, including Ras-associated proteins 6, 7, 9, and 11 (Rab6, Rab7, Rab9, and Rab11), early endosome antigen 1 (EEA1), lysobisphosphatidic acid (LBPA), and lysosomal-associated membrane protein 1 (LAMP1), which can be utilized to monitor the progression of ONs through the endosomal pathway and identify potential points of endosomal escape (Created with BioRender: TQ25720MWN).

**Figure 5 pharmaceutics-15-01130-f005:**
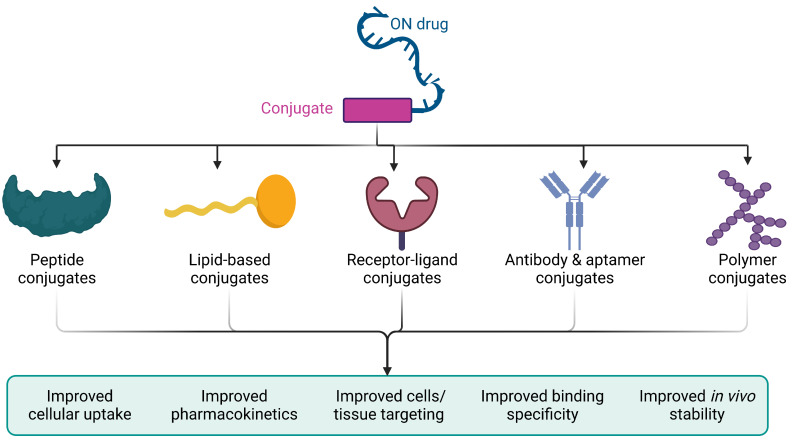
ON bioconjugation. Bioconjugation improves the cellular uptake, pharmacokinetics, targeting, and binding specificity of ON, leading to improved stability, efficacy, and safety (Created with BioRender: ZJ25720INP).

**Table 1 pharmaceutics-15-01130-t001:** **Comparison of different ON platforms.** The binding efficiency, tissue/cellular target, uptake level, off-target effects, liver toxicity, and immunogenicity of each platform can vary depending on the specific ON strategy used, the target RNA sequence, and other factors [11,29,39,60,103,104,105,106,107,108]. The information presented in this table are based on a general understanding of data from reported records and may vary depending on a case-by-case basis.

Platform	Mechanism	Most Used ON Chemistry	Length (nt)	Primary Cellular/Tissue Target	Administration Route	Uptake Level	Off-Target Effects	Major Undesirable Effects
Gapmers	RNase H-mediated cleavage	PS, LNA, 2′OMe, 2′MOE	12–30	Cytoplasm, nucleus	SC, IV, IP	Moderate to high; high in liver, variable in other tissues depending on delivery method	Low with appropriate design and modifications	Variable hepatotoxicity depending on chemistry and dose, potential for immunogenicity
Aptamers	Binding and inhibiting target proteins via shape complementarity	RNA, DNA	20–100	Extracellular space, cell surface proteins	SC, IV, IP, oral	Low; high in plasma, low in tissues due to size and charge barriers	Low due to high specificity and affinity of binding	Rare hepatotoxicity, variable potential for immunogenicity, bleeding risk, rapid renal clearance
Steric blocking ONs	Binding and blocking target mRNA/pre-mRNA	PS, 2′Ome, 2′MOE, LNA, PMO, PNA	8–50	Cytoplasm, nucleus	SC, IV, IP, topical, oral, rectal	Variable depending on target accessibility and delivery vehicle	Low with appropriate design and modifications	Variable hepatotoxicity depending on chemistry and dose, low potential for immunogenicity, injection site reactions, RNase-activation
Immunostimulant ONs	Activating innate immune cells via TLR recognition of unmethylated CpG motif sequences	CpG DNA, modified RNA	16–28	Toll-like receptor, dendritic cells, B cells	SC, IV, intratumoral	Low to high; high in immune cells expressing TLR9, low in other tissues	Low due to specific TLR recognition of CpG motifs	Rare hepatotoxicity, high potential for immunogenicity, cytokine induction, injection site reactions
Antagomirs	Binding and inhibiting endogenous miRNAs	2′OMe, 2′MOE, 2′-F, LNA, PNA, PMO	15–31	Cytoplasm, endoplasmic reticulum, nucleus, and extracellular space	SC, IV	High; high in liver, variable in other tissues depending on delivery method	Low due to high specificity and affinity of binding	Variable hepatotoxicity depending on chemistry and dose, low potential for immunogenicity, bleeding risk, rapid renal clearance
Agomirs	Mimicking and enhancing endogenous miRNA activity	2′OMe, 2′MOE, 2′-F, LNA, PNA, PMO	19–23	Cytoplasm, endoplasmic reticulum, nucleus, and extracellular space	SC, IV	High; variable depending on target accessibility and delivery methods	Low due to high specificity and affinity of binding	Variable hepatotoxicity depending on chemistry and dose, low potential for immunogenicity
siRNA	RNAi-mediated gene silencing	RNA, PS, 2′OMe, 2′MOE, 2′-F, LNA	18–24	Cytoplasm	SC, IV, topical	High; high in liver, variable in other tissues depending on delivery method	Low with appropriate design and modifications; sequence-specific RNAi	Rare hepatotoxicity, variable potential for immunogenicity, off-target gene deactivation

PS, phosphorothioate; 2′MOE, 2′-O-methoxyethyl; 2′OMe, 2′-O-methy; LNA, locked nucleic acid; RNA, ribonucleic acid; DNA, deoxyribonucleic acid, PNA, peptide nucleic acid; SC, subcutaneous; IV, intravenous; IP, intraperitoneal.

**Table 2 pharmaceutics-15-01130-t002:** ON delivery systems. This outlines the general advantages and disadvantages of each delivery system and its potential for use with each ON platform.

ON Delivery System	Advantages	Limitations
Lipoplex, liposomes, and lipid nanoparticles	Biocompatible and versatile delivery vehicle Protection from enzymatic degradation Enhanced stability	Potential for immunogenicity and toxicity Limited tissue specificity Susceptible to degradation and may suffer from leakage of the encapsulated drug
Peptide-based delivery systems	High specificity and efficacy Biodegradable Can target specific types of cells	Challenges in large-scale production Stability concerns due to protease sensitivity May be subject to proteolytic degradation
Polymer-based delivery systems	Versatility in structure and function Improved stability and solubility High loading capacity and controlled release	Potential for immunogenicity and toxicity Limited tissue specificity May be subject to degradation and clearance by the body
Antibody complexation delivery systems	High specificity and affinity for targets Enhanced cellular uptake Ability to cross biological barriers	Limited delivery efficiency Immunogenicity concerns Production is expensive and sophisticated
Extracellular vesicle-mediated delivery systems	Natural carriers with high stability and low immunogenicity Low immunogenic potential	Challenges in isolation, purification, and large-scale production Limited loading capacity
Spherical nucleic acids	Unique and highly efficient delivery mechanism Resistance to enzymatic degradation Ability to cross biological barriers	Challenges in large-scale production Limited tissue specificity
DNA nanostructures	Programmable and predictable structures Multifunctionality and versatility Ability to incorporate various therapeutic and targeting moieties	Stability concerns in biological environments Challenges in large-scale production
Stimuli-responsive delivery technologies	Controlled release in response to external or internal stimuli Enhanced targeting specificity Potential for reduced toxicity and side effects	Complexity in design and fabrication Potential off-target effects due to stimuli-responsive components
Multilayered biopolymer nanofilms	Controlled release Capability to incorporate large biomacromolecules Potential for spatial and temporal control of nucleic acid delivery	Complexity in fabrication Limited diffusion of large biological polyelectrolytes via multilayered assemblies Further research required for optimization

## Data Availability

Not applicable.

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
