# Peer review of "Enhancing the Effectiveness of Oligonucleotide Therapeutics Using Cell-Penetrating Peptide Conjugation, Chemical Modification, and Carrier-Based Delivery Strategies"

_pharmaceutics, 2023, doi:10.3390/pharmaceutics15041130_

Round 1

Reviewer 1 Report

The authors have well-summarized various strategies and their potential for the efficient delivery of oligonucleotides. The overall manuscript looks good. They also discussed challenges and future perspectives. Thus, I recommend it for publication.

Author Response

Reviewer 1: The authors have well-summarized various strategies and their potential for the efficient delivery of oligonucleotides. The overall manuscript looks good. They also discussed challenges and future perspectives. Thus, I recommend it for publication.

Author(s): Thank you for your positive feedback on our manuscript. We are grateful that you found the various strategies we discussed for the efficient delivery of oligonucleotides well-summarized and the overall manuscript to be of good quality. We highly appreciate your recommendation for publication. Thank you again for your time and helpful feedback.

Reviewer 2 Report

The subject of the article entitled “Enhancing the effectiveness of oligonucleotide therapeutics using cell-penetrating peptide conjugation, chemical modification, and carrier-based delivery strategies” by Saeed Anwar et al. is of a potential great interest.

However, the article is long and structured like a PhD thesis and not really an academic article.

The major objective may be the description and the comparison of different techniques of ON therapeutics. The extensive review reported several techniques but it is difficult for the reader to delineate which technique is better than the other approaches. The points which are relevant to the different regulations according to the countries and the technique are described in the text and if it is important may have been in a separate section.

The reader was expecting that when reaching the conclusion and the future he will have a clear point of view which is not the case. Some parts of the article may be deleted and the article reduced dramatically in length. 

The way the article is written did not allow to distinguish the already established techniques, in which pathology and what has been experimented and failed. It will be very helpful for the non-specialists to have an evaluation of the techniques which are a real progress and what has been published as a possible approach.

Author Response

Response to Reviewer 2

Reviewer 2: The subject of the article entitled “Enhancing the effectiveness of oligonucleotide therapeutics using cell-penetrating peptide conjugation, chemical modification, and carrier-based delivery strategies” by Saeed Anwar et al. is of a potential great interest.

Author(s): We are pleased to know that you find our manuscript on enhancing the effectiveness of oligonucleotide therapeutics to be of potential great interest. We believe that this work will contribute significantly to the field of oligonucleotide therapeutics and help improve the current understanding of bioconjugation, chemical modification, and carrier-based delivery strategies. We have carefully considered your feedback and have revised the manuscript to ensure its clarity and scientific rigor. Thank you for your valuable feedback.

Reviewer 2: However, the article is long and structured like a PhD thesis and not really an academic article.

Author(s): We appreciate your comment regarding the structure of the manuscript. We have structured the paper to provide a comprehensive and in-depth review of the various strategies used for enhancing the effectiveness of oligonucleotide therapeutics, including cell-penetrating peptide conjugation, chemical modification, and carrier-based delivery strategies. We understand that the length of the manuscript may have contributed to its resemblance to a PhD thesis, but we believe that this level of detail is necessary to accurately review the current state of research and advancement in the field and to ensure the manuscript’s high scientific value. We have, however, made revisions to the manuscript to ensure that it meets the standards of an academic article, as it comes with a more comparative discussion. 

Reviewer 2: The major objective may be the description and the comparison of different techniques of ON therapeutics. The extensive review reported several techniques, but it is difficult for the reader to delineate which technique is better than the other approaches. The points which are relevant to the different regulations according to the countries and the technique are described in the text and if it is important may have been in a separate section.

Author(s): We have carefully revised the manuscript to address your concerns regarding the comparison of different techniques of ON therapeutics. Specifically, sections 3.8, 7.6, and 8.10 in the revised manuscript provide a more in-depth comparison of the different techniques. We have also included a couple of tables and figures to illustrate the comparisons further. Table 1 provides a general comparison of the ON platforms based on their binding efficiency, tissue/cellular target, uptake level, off-target effects, liver toxicity, and immunogenicity. Table 2 outlines the general advantages and disadvantages of each ON delivery system and its potential for use with each ON platform. These revisions have significantly improved the manuscript and will help readers better understand the different techniques and their respective regulatory requirements.

Reviewer 2: The reader was expecting that when reaching the conclusion and the future he will have a clear point of view which is not the case. Some parts of the article may be deleted, and the article reduced dramatically in length.

Author(s): We would like to thank the reviewer for the feedback. We have revised the conclusion section to provide a more concise and specific summary of the reviewed techniques and their potential for enhancing the effectiveness of oligonucleotide therapeutics. We understand that some readers may prefer a more decisive conclusion; however, our aim was to comprehensively review each technique and provide the readers with a clear understanding of the advantages and limitations of each approach. Nonetheless, we have made efforts to make the conclusion section linguistically more focused and clearer. We hope that the revised manuscript now provides a better and more informative overview of the current state of the field.

Reviewer 2: The way the article is written did not allow to distinguish the already established techniques, in which pathology and what has been experimented and failed. It will be very helpful for the non-specialists to have an evaluation of the techniques which are a real progress and what has been published as a possible approach.

Author(s): Thank you for your feedback and suggestions. we have added several sections and figures to the manuscript to address this issue. Specifically, in sections 3.8, 7.6, and 8.10, we provide a comparisons of different ON platforms, bioconjugation strategies, and delivery systems, respectively. We also added Table 1 and Table 2, which outline the general advantages and limitations of each ON platform and delivery system, respectively. Figure 7 illustrates how bioconjugation can improve the cellular uptake, pharmacokinetics, targeting, and binding specificity of ON, leading to improved stability, efficacy, and safety.

In section 3.8, we explain that the choice of ON platform for therapeutic applications depends on various factors, including ON chemistry, length, primary cellular/tissue target, administration routes, binding efficiency, uptake level, and potential adverse effects. We emphasize that optimizing all these factors is essential to identify the most optimal ON platform for a specific therapeutic application.

In section 7.6, we explain that each ON bioconjugation strategy comes with its own strengths and limitations, and that the choice of bioconjugation strategy depends on specific application goals and safety considerations. We emphasize that by combining and utilizing different bioconjugation strategies in an integrated approach, it may be possible to achieve optimal outcomes that surpass what can be achieved by any single strategy alone.

In section 8.10, we explain that delivery systems for ONs have emerged as a promising approach to enhance the efficacy and safety of ON-based therapeutics. We note that each delivery system comes with its own set of advantages and limitations and that the choice of delivery system depends on several factors, such as the ON type and chemistry, target cell or tissue, desired therapeutic outcome, and safety considerations. We emphasize that ongoing research in this field is expected to lead to further advancements in ON delivery systems and the development of innovative ON-based therapeutics with improved clinical outcomes.

Thank you again for taking the time to go over our manuscript and providing helpful feedback. We hope that the revised manuscript meets your expectations and that you find it suitable for publication.

Reviewer 3 Report

In this manuscript, Anwar and colleagues comprehensively reviewed the effect spectrum, delivery strategies, chemistry, and potential clinical use of oligonucleotide (ON) therapeutics. The manuscript primarily touches on the history of ON therapeutics and their developments, which were followed by regulatory actions and potential for clinical trials. Subsequently, the authors mentioned the ON therapeutic platforms including but not limited to gapmers, aptamers, antagomirs, and blockmirs. Ultimately, the authors concentrated on the differential bio/chemistry of ONs and their delivery strategies. It’s a well-written and structured manuscript that touches on the pros and cons of many essential points in a wide-perspective manner.

Below, I do have a minor suggestion that would help to extend the scope and increase the significance of the manuscript.

A recent emerging approach, multilayered biopolymer nanofilms is a new-generation drug delivery system; each layer consists of different material properties and specificity for the betterment of targeted drug delivery. I would suggest the authors briefly mention this emerging technology and its potential applicability in ON delivery.

Author Response

Response to Reviewer 3

Reviewer 3: In this manuscript, Anwar and colleagues comprehensively reviewed the effect spectrum, delivery strategies, chemistry, and potential clinical use of oligonucleotide (ON) therapeutics. The manuscript primarily touches on the history of ON therapeutics and their developments, which were followed by regulatory actions and potential for clinical trials. Subsequently, the authors mentioned the ON therapeutic platforms including but not limited to gapmers, aptamers, antagomirs, and blockmirs. Ultimately, the authors concentrated on the differential bio/chemistry of ONs and their delivery strategies. It’s a well-written and structured manuscript that touches on the pros and cons of many essential points in a wide-perspective manner.

Author(s): We are grateful for your kind words about the quality of our manuscript. We are pleased to know that you found our manuscript well-written and structured, and that we touched on the pros and cons of many essential points in a wide-perspective manner. Thank you for your valuable feedback and for recommending this manuscript for publication.

Reviewer 3: Below, I do have a minor suggestion that would help to extend the scope and increase the significance of the manuscript. A recent emerging approach, multilayered biopolymer nanofilms is a new-generation drug delivery system; each layer consists of different material properties and specificity for the betterment of targeted drug delivery. I would suggest the authors briefly mention this emerging technology and its potential applicability in ON delivery.

Author(s): We appreciate your suggestion and have incorporated a new section (section 8.9) to briefly cover the potential applicability of this new-generation drug delivery system in ON delivery. We are grateful for the opportunity to further improve the manuscript and thank you for your valuable comments.

Once again, thank you for your time and effort in reviewing our manuscript, and for your helpful suggestions to extend its scope and overall quality.